# Deep Neural Networks for Road Sign Detection and Embedded Modeling Using Oblique Aerial Images

Zhu Mao [1,†], Fan Zhang [1], Xianfeng Huang [1,*,†], Xiangyang Jia [1], Yiping Gong [1] and Qin Zou [2]

1 State Key Laboratory of Information Engineering in Surveying, Mapping and Remote Sensing, Wuhan University, 129 Luoyu Road, Wuhan 430079, China; maoz@whu.edu.cn (Z.M.); zhangfan@whu.edu.cn (F.Z.); jiaxiangyang@whu.edu.cn (X.J.); gongyp15@163.com (Y.G.)
2 School of Computer Science, Wuhan University, Wuhan 430072, China; qzou@whu.edu.cn
* Correspondence: hwangxf@gmail.com
† These authors contributed equally to this work.

**Abstract:** Oblique photogrammetry-based three-dimensional (3D) urban models are widely used for smart cities. In 3D urban models, road signs are small but provide valuable information for navigation. However, due to the problems of sliced shape features, blurred texture and high incline angles, road signs cannot be fully reconstructed in oblique photogrammetry, even with state-of-the-art algorithms. The poor reconstruction of road signs commonly leads to less informative guidance and unsatisfactory visual appearance. In this paper, we present a pipeline for embedding road sign models based on deep convolutional neural networks (CNNs). First, we present an end-to-end balanced-learning framework for small object detection that takes advantage of the region-based CNN and a data synthesis strategy. Second, under the geometric constraints placed by the bounding boxes, we use the scale-invariant feature transform (SIFT) to extract the corresponding points on the road signs. Third, we obtain the coarse location of a single road sign by triangulating the corresponding points and refine the location via outlier removal. Least-squares fitting is then applied to the refined point cloud to fit a plane for orientation prediction. Finally, we replace the road signs with computer-aided design models in the 3D urban scene with the predicted location and orientation. The experimental results show that the proposed method achieves a high mAP in road sign detection and produces visually plausible embedded results, which demonstrates its effectiveness for road sign modeling in oblique photogrammetry-based 3D scene reconstruction.

**Keywords:** embedded modeling; photogrammetry-based 3D city model; oblique aerial image; stereo vision; road sign detection; small object detection



## 1. Introduction

Real-world three-dimensional (3D) urban models are important in building "smart cities" and supporting numerous applications such as city planning, space management, and intelligent traffic systems [1]. In recent years, with the development of unmanned aerial vehicles (UAVs), oblique photogrammetry has been widely used to create 3D urban models because it collects abundant information on a large scale with the benefit of low cost and high efficiency [2,3].

However, due to slice- and pole-like shape features, weak texture and high camera incline angle, oblique photogrammetry-based 3D modeling of some artifacts, such as light poles and road signs, remains challenging. Road signs, which play crucial roles in city infrastructure, are set up at the sides of roads and artificially designed with striking colors and regular sliced shapes to provide navigation information and warnings to drivers and pedestrians [4,5]. Reconstructed road signs are fragmentary with discontinuous surfaces and blurred textures in the 3D scene, exposing the defects of oblique photogrammetry-based methods.

Cases of 3D road sign models reconstructed via an aerial photogrammetry-based approach are shown in Figure 1. We consider the reasons for defects from two perspectives.

(1) The limited number of 3D points is not sufficient to generate complete models of road signs with continuous surfaces due to image resolution and the accuracy of feature matching algorithms. Low-quality oblique aerial images also produce a blurred texture of road signs.

(2) The sliced shape road signs are too thin to distinguish both sides. Thus, the cloud points are merged into a whole part and cannot be separated easily for meshing.

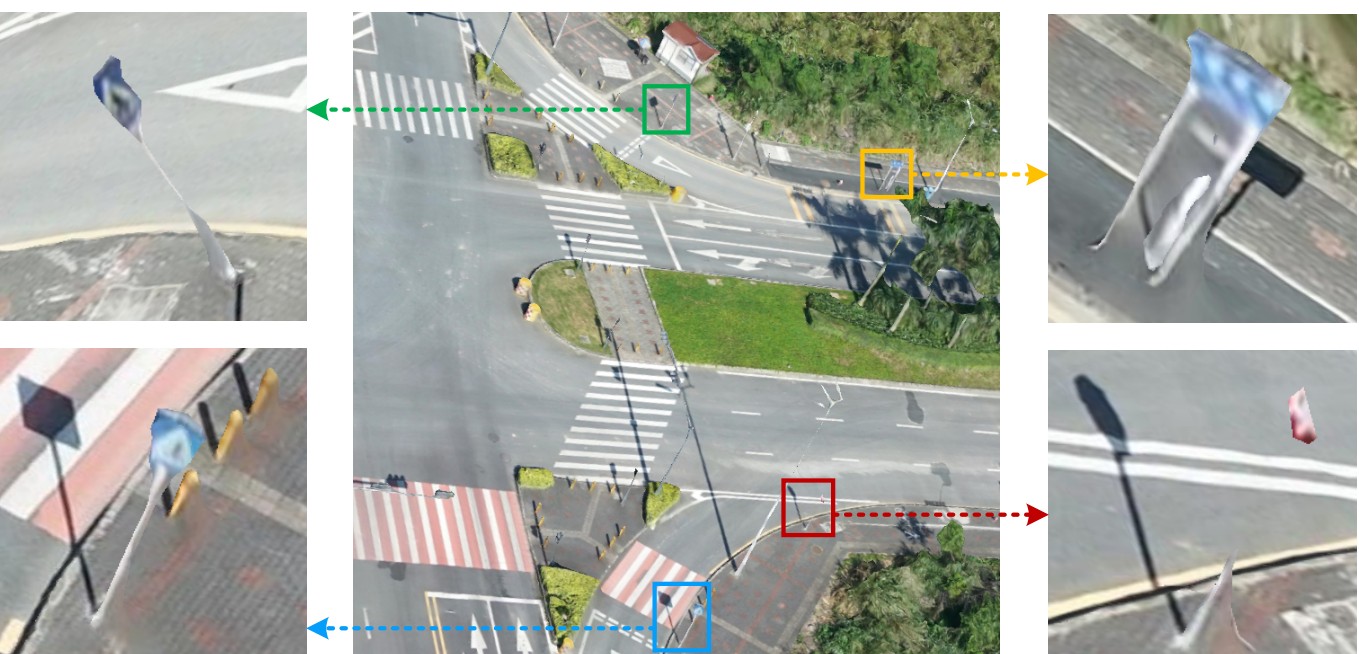

**Figure 1.** Examples of road sign models reconstructed via the software ContextCapture in urban scenes.

In oblique aerial image-based 3D city models, large-scale incomplete road signs commonly lead to less informative guidance and unsatisfactory appearance. Generally, fragmentary road signs are usually erased manually to embellish 3D urban models. The task of model optimization is undoubtedly labor intensive and time consuming and requires prior knowledge. As 3D city models are popularly used in various applications, road signs augment the visual realism and enrich traffic information of the reconstructed city models. In addition to their significance in generating road databases, road signs provide valuable information for navigation in 3D scenes; thus, the modeling of road signs is essential.

Additionally, the overwhelmingly popular deep CNN provides smart detectors to locate and recognize road signs in massive oblique aerial images with high performance. Therefore, one alternative is to leverage the detected 2D location and semantic information to localize road signs in a 3D scene via triangulation. Moreover, it is conceivable to fit a 3D bounding box and a plane that estimates the 3D location and orientation of road signs using the triangulation result.

Compared to existing object detectors, region-based CNN methods perform supremely well in object detection tasks with high localization and recognition accuracy [6–9]. However, the gap in the performance between the detection of small and large objects is significant [10]. Small object detection is challenging since small objects do not contain detailed information and may even disappear in the deep network. In addition, defects such as complex training procedures with many heuristics and hyperparameters [11], imbalances during the training process [12] and predefined dense anchoring schemes where anchors are sampled uniformly over the spatial domain with a predefined set of scales

and aspect ratios [13] still exist. On the other hand, compared to generic images, oblique aerial images are taken with high-tilting camera angles that lead to variable deformation of objects. Generally, road signs are sparsely distributed in oblique aerial images with blurred texture and a variable shooting view. Thus, detecting road signs from oblique aerial images is challenging.

To address the road sign modeling issue, this paper presents a pipeline for embedding road sign models based on a deep CNN using oblique aerial images. The proposed method also provides a referenced solution for the unresolved reconstruction problem of some characteristic objects, such as street lamps. The framework of our approach includes road sign detection, localization, orientation, and computer-aided model embedding.

Related works on oblique photogrammetry-based modeling, 3D scene augmentation, road sign detection and small object detection are discussed in Section 2. The details of our proposed approach and its key steps are described in Section 3, which effectively promotes road sign modeling tasks. The proposed approach begins by recognizing road signs for location and classification using a region-based CNN and a data synthesis strategy. We then classify road signs into three categories based on functionality: warning, mandatory and prohibitory. Detecting road signs from oblique aerial imagery is challenging due to the infinite number of background objects. Moreover, road signs do not frequently appear in each oblique aerial image containing them, which depresses the detection performance. Second, under the geometric constraints placed by the bounding boxes, we use the SIFT [14] feature to extract the corresponding points on the road signs. Third, we obtain the coarse location of a single road sign by triangulating the corresponding points and refine the location via outlier removal. Least-squares fitting is then applied to the refined point cloud to fit a plane for orientation prediction. Finally, we replace the road signs with computer-aided design models in the 3D urban scene based on the predicted location and orientation. Experiments are introduced and analyzed in Section 4, and Sections 5 and 6 present the discussion and conclusion, respectively.

## 2. Related Work

### 2.1. Oblique Photogrammetry-Based Modeling and 3D Scene Augmentation

Oblique photogrammetry, a cost-effective and flexible data acquisition approach, provides a new data source with distinct advantages: detailed information about facades, multiple views from various perspectives and a substantially varying image scale [15–17]. The growing interest of the scientific and industrial community and software developers in using oblique aerial images to build 3D models has made the advantages of the technique evident [18,19]. Even though remarkable progress has been made in recent years, oblique aerial image-based 3D modeling may have defects caused by the occlusion and large tilting angles of the camera. However, defects are exceptionally prominent for small and sliced shape objects.

Methods have been proposed to optimize oblique photogrammetry-based 3D models. Integration of aerial and ground images [20,21] is beneficial to enhance the surface reconstruction in urban environments. Additional approaches [22–26] have been developed to augment 3D building models. However, most of them focus on building façades and surface optimization via feature matching in urban areas but not in the context of street objects, such as road signs.

Additionally, studies exploit matching-based [27,28] and scene synthesis methods [29,30] to optimize the reconstruction results and retrieve and align 3D models from an extensive database. However, most of these approaches focus on indoor environment reconstruction based on RGB-D scans or LiDAR data. [31] embedded user-generated content to reduce the complexity of the oblique photogrammetry-based 3D model, yet they manually delineated the boundary for scene editing and ignored some essential artifacts such as road signs.

## *2.2. Road Sign Detection*

In recent years, CNN-based object detection has experienced impressive progress. Comprehensive surveys [32–35] have systematically analyzed and discussed the advances, applications and pending issues related to object detection. State-of-the-art algorithms for road sign recognition and tracking are used to support driver-assistance systems (ADAS) [32–35], autonomous vehicles [5,36,37], and road safety inspections. Most of them detect road signs from street-view images, yet few of them concentrate on 3D reconstruction or detection from oblique aerial images.

In addition, various approaches [38,39] have been proposed to manage the task of road sign 3D localization using panoramic images and depth-map data collected via a mobile mapping system or RGB images to estimate the 3D location. Ref. [40] proposed a pipeline for 3D reconstruction of traffic signs via detection, shape fitting and template matching from multistreet view images, where the silhouette and texture of traffic signs are crystal clear. In general, few studies have been conducted on the detection or modeling of road signs through oblique aerial images.

Compared to street-view data, panoramic images, or satellite imagery, road signs are relatively small with blurred texture and unpredictable distortion in oblique aerial images due to the limitations of image resolution and varying shooting angles. Moreover, road signs do not appear frequently even in oblique aerial images containing them, resulting in a major imbalance in the dataset. As a result, it is more challenging to detect road signs from oblique aerial images.

## *2.3. Small Object Detection with Balanced Learning and Guided Anchoring*

To manage small object detection tasks, multiscale feature learning [41,42], data augmentation, training strategy [43,44], context-based detection, improved loss function [45] and GAN-based [46] detection methods have been applied to enhance the performance. Moreover, some studies [47–49] have detected small objects from satellite remote sensing imagery, but detection studies based on oblique aerial images are rare.

For imbalance problems [50], oversampling and stitching data augmentation [51,52] and effective frameworks [12,53] are leveraged to decrease the negative effect of category imbalance in the training dataset and training process, respectively. Regarding data synthesis, most approaches cut object masks from the training data first and then render them on background imagery directly for data augmentation.

The width-to-height ratios of road signs in oblique aerial images are not fixed since the high tilting camera angle leads to variable distortion. In this case, the dense and uniform anchoring scheme is not necessarily the optimal way to prepare the anchors for road sign detection. Several approaches [13,54,55] have proven that anchors can be sufficiently intelligent to generate in proper scale and location by themselves with the advantage of saving effort and improving accuracy, and there is no need to manually predefine the sophisticated dense anchoring scheme.

## 3. Method

Road signs augment the realism and semantic information of 3D urban scenarios, yet it is difficult to reconstruct them via oblique photogrammetry. Meanwhile, deep learning algorithms provide an alternative to locate and classify road signs from massive images with high performance. Thus, this paper proposes an embedded modeling method based on a region-based CNN to address this issue.

In this section, we introduce the overall framework of the proposed method for road sign embedded modeling in oblique photogrammetry-based 3D scenarios. Figure 2 depicts the overall structure of the proposed framework. The main steps are as follows:

(1) Data synthesis and road sign detection. We present an end-to-end balance-learning framework for small object detection that takes advantage of the region-based CNN and a data synthesis strategy. First, data synthesis and augmentation are applied to moderate the negative effects caused by object imbalance in the training dataset. Second,

a region-based CNN that combines balanced learning with guided anchoring strategies is used to enhance the detection results, as illustrated by comparative experiments.

(2) Road sign 3D localization and orientation. Stereo vision of multiple image sets is used for 3D localization and orientation. First, all oblique aerial images containing the same road signs are grouped according to the imaging similarity. Second, under the geometric constraints of the bounding boxes, we use the SIFT feature to extract the corresponding points on the road signs of an image set. Third, we apply triangulation to the corresponding points in each image group to obtain 3D points. The triangulation results from all image groups are merged to generate the approximate location of a single 3D road sign. Additionally, we remove sparse outliers through statistical analysis to refine the 3D point cloud for a more precise 3D location. Finally, least-squares fitting is applied to the refined point cloud to fit a plane for orientation prediction. The fitted plane should be vertical to the known surface of the street and indicate the orientation of a road sign.

(3) Model matching and embedding. On the basis of the classification results, we retrieve computer-aided design models of road signs from the database via template and texture matching and then embed them in 3D urban scenes with the predicted location and orientation.

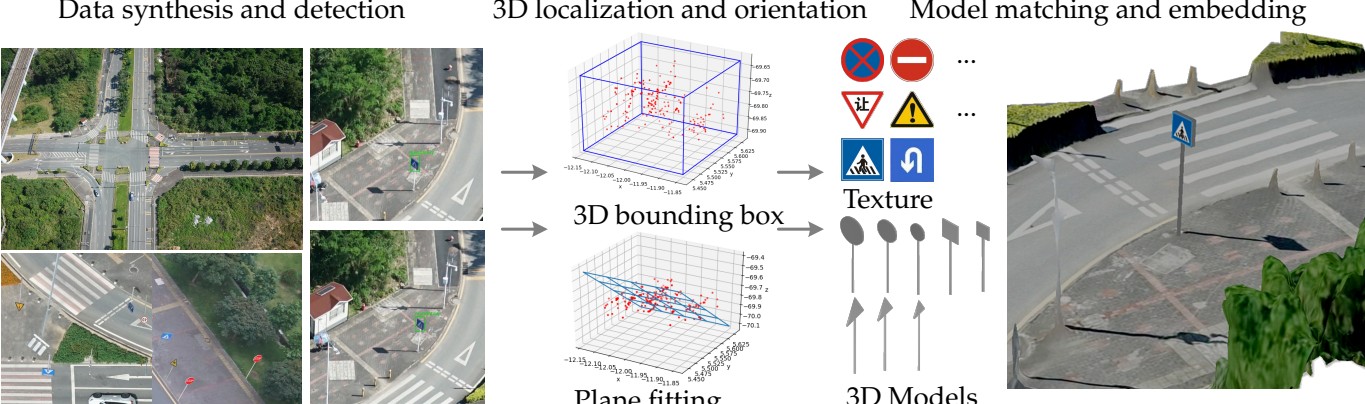

**Figure 2.** Overall framework of the proposed embedded modeling method.

### 3.1. Road Sign Detection

Road signs are difficult to detect from oblique aerial images of the complicated real world. There are three main challenges. (1) Road signs are usually small in terms of pixel size with blurred textures due to the limitation of image resolution, which makes small objects more difficult to detect than large ones. (2) Road signs do not appear frequently, even in images that contain them, resulting in class imbalance in the training dataset. (3) The distortion caused by high shooting angles increases the interclass differences, making it difficult to distinguish real road signs and fake targets in complex street scenes. Concerning image resolution limits, it is not easy to improve the quality of oblique aerial images immediately, as it depends significantly on the data collection equipment.

Therefore, these challenges can be summarized as an imbalance problem in the training dataset and during the training process. In our method, we leverage a balanced learning strategy to alleviate the negative effect of the imbalance problem. Additionally, we adopt a guided anchoring strategy to eliminate complicated manual anchor design and thus avoid uncertain impacts it may bring. Generally, we present an end-to-end balance-learning framework for small object detection that takes advantage of the region-based CNN and a data synthesis strategy (see Figure 3). The detection pipeline takes Faster R-CNN [7] as the basis and Resnet101 [56] as the basic feature extractor.

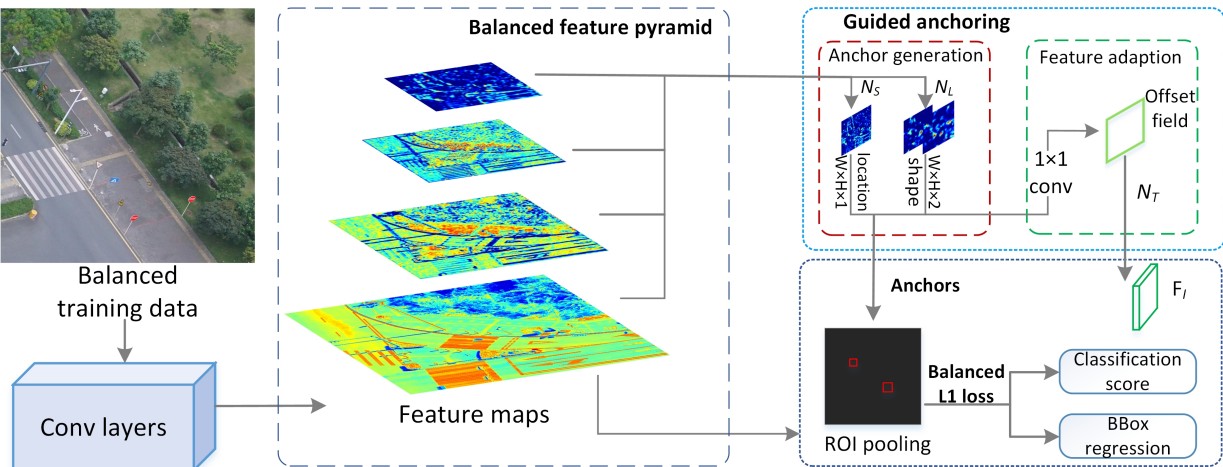

**Figure 3.** The end-to-end balance-learning framework for small object detection takes advantage of the region-based CNN and a data synthesis strategy.

In this study, we split the imbalance problems into two types: that in the training dataset and the during the training procedure. It is crucial to alleviate the imbalance to achieve optimal training and fully exploit the potential of model architectures. Specifically, data synthesis and augmentation and training techniques, including IoU-balanced sampling, balanced feature pyramid, and balanced L1 loss, are used to mitigate the adverse effects of the aforementioned imbalance. In addition, to reduce efforts for sophisticated anchor settings and design, a guided anchoring strategy [13] is adopted in our research, avoiding the negative and uncertain impacts on detection performance caused by unsuitable anchor settings. Much higher quality region proposals are generated using the guided anchoring method, thus achieving an enhancement in accuracy.

### 3.1.1. Imbalance Problems in the Training Dataset

Due to the high and tilting shooting angle, road signs are small in terms of pixel size, sparsely distributed and distorted in oblique aerial images. Two issues in the training dataset result in object imbalance. (1) Each image contains only a few road signs and a large ratio of background. (2) Interclass differences exist since the orientation and brightness of road signs in oblique aerial images vary greatly. The imbalance of objects in the training images results in overlap between small ground-truth objects, and the predicted anchors are much lower than the expected IoU threshold. The primary purpose of synthesizing data is to balance the number of oblique aerial images and objects to address the imbalance problem in the training dataset. To better learn the distribution of our training dataset, the quantity of each road sign category in the oblique aerial images is shown in Table 1. There are 450 images with a pixel size of $1024 \times 1024$.

**Table 1.** The number of oblique aerial images and objects of each road sign category.

| Class | Mandatory Sign | Warning Sign | Prohibitory Sign |
|---|---|---|---|
| Image number | 358 | 63 | 131 |
| Object number | 509 | 71 | 227 |

The above statistical results indicate considerable category imbalance in the training dataset, which adversely impacts the detection results of road signs with a small number of samples. For example, the number of mandatory signs is nearly six times that of warning signs, while the number of mandatory signs is more than seven times that of warning signs. In addition, road signs occupy fewer image areas, and the less frequent road signs usually

have fewer matched anchors, which may make it more difficult to learn useful information from the network.

In contrast to simply cutting and pasting to synthesize data [57], the proposed data synthesis and augmentation method is designed based on computer-aided design 3D road sign models to balance the quantity of each road sign category and to reduce the negative impact of the imbalance in the training dataset. Furthermore, the transformation of 3D rotation and lighting conditions are taken into account to make the synthetic data more realistic. The proposed method for data synthesis and augmentation is illustrated in Figure 4.

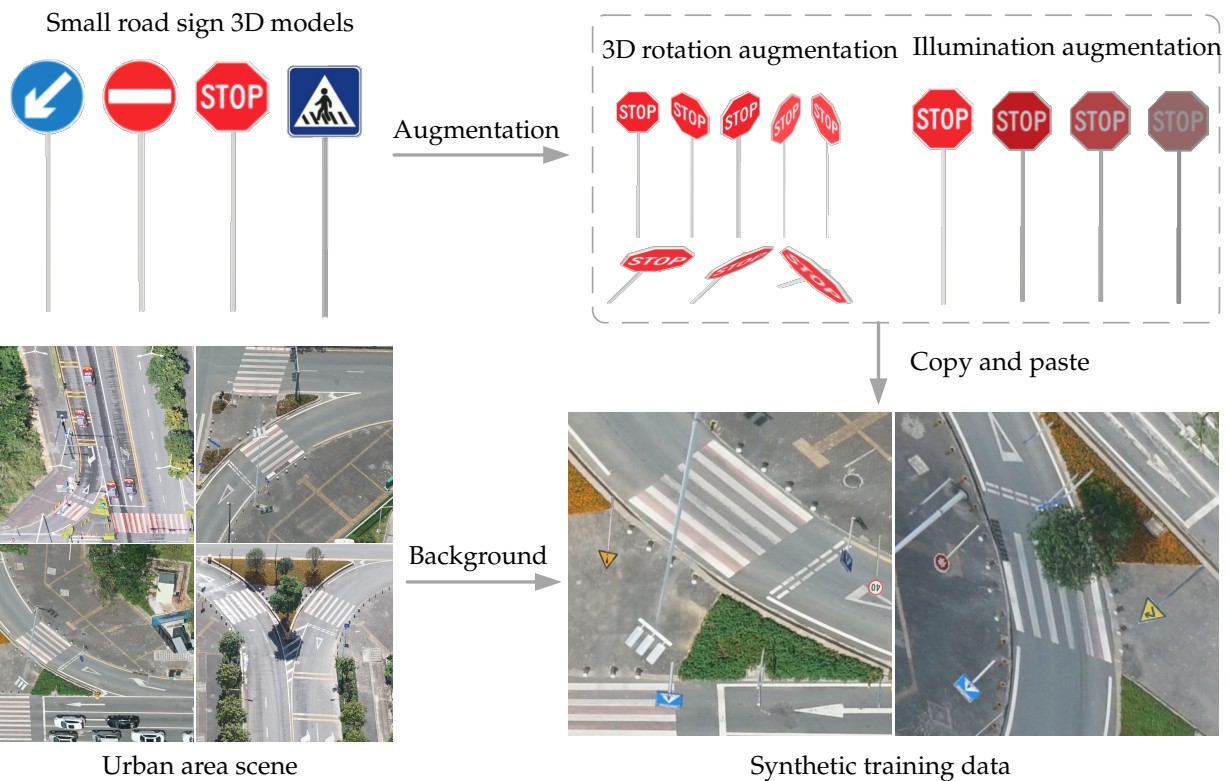

**Figure 4.** Schematic of data synthesis and augmentation with 3D rotation and lighting transformation using computer-aided design 3D road sign models.

The main steps of data synthesis and augmentation are as follows:

(1) A collection of computer-aided design 3D road sign models is created via the 3D modeling software 3DSMax. Considering the lighting conditions, we rotate the 3D road sign models in three dimensions to generate road sign masks.

(2) According to the statistical information on the number of road signs within each category in the original training dataset, we render the road sign mask, which is scaled to the appropriate size on random background images, to create synthetic training images.

(3) We count the number of signs in each category in the synthetic training images to keep the number of oblique aerial images and objects in the training dataset as balanced as possible.

(4) We augment the synthetic data via rotation, flip and gamma transformations to avoid overfitting and to improve the detection accuracy.

### 3.1.2. Imbalance Problems in the Training Process

During the training process, the selected region samples, the extracted visual features and the designed objective function have significant impacts on the performance

of the detection model. The imbalance issues that exist in these three aspects prevent the power of well-designed model architectures from being fully exploited, thus limiting overall performance. Referring to the Libra R-CNN [12], we apply IoU-balanced sampling, balanced feature pyramid, and balanced L1 loss to mitigate the imbalance during the training process.

Specifically, in contrast to random sampling, the IoU-balanced sampling strategy evenly splits the sampling interval into K bins according to the IoU, the N demands are equally distributed in each bin, and training samples are selected from uniformly from the bins.

$$p_k = \frac{N}{k} * \frac{1}{M_k}, k \in [0, L), \tag{1}$$

where $M_k$ is the number of sampling candidates in the corresponding interval denoted by $k$. Experiments show that the performance of this approach is not sensitive to $k$.

For the imbalance between low-level and high-level features, enhanced multilevel features using the same deeply integrated balanced semantic feature are applied, generated through rescaling, integrating, refining, and strengthening processes based on original feature maps. After rescaling the multilevel features to the same size, the balanced semantic features are obtained by simple averaging

$$C = \frac{1}{L} \sum_{l=l_{min}}^{l_{max}} C_l, \tag{2}$$

where $L$ is the number of multilevel features and $l_{min}$ and $l_{max}$ represent the lowest and highest levels, respectively. Then, the balanced semantic features are further refined for detection.

To balance the classification and localization tasks in detection, we use the balanced $L_1$ loss to balance the involved tasks by tuning their loss weights. The balanced $L_1$ loss function is provided in Equation (3).

$$L_b(x) = \begin{cases} \frac{\alpha}{b}(b|x| + 1)ln(b|x| + 1) - \alpha|x| & \text{if } |x| < 1 \\ \gamma|x| + C & otherwise, \end{cases} \tag{3}$$

in which the parameters $\gamma$, $\alpha$ and b are constrained by

$$\alpha ln(b + 1) = \gamma \tag{4}$$

### 3.2. Road Sign 3D Localization and Orientation

Based on the detection result, multiple oblique aerial images are used to retrieve the 3D position and orientation of road signs, each acquired from a different point of view in space. Three main steps are involved in the process of 3D localization and orientation.

(1) Group all oblique aerial images containing the same road signs according to imaging conditions, such as shooting angles and lighting conditions.
(2) Search corresponding points in image groups. Under the geometric constraints placed by the bounding boxes, the SIFT feature and optimized brute-force matcher are applied to extract corresponding points.
(3) Apply corresponding points for triangulation to obtain a coarse location and refine it via outlier removal. Then, merge 3D points of the same road signs to obtain the location of a single 3D road sign. Finally, we fit a plane to the refined point cloud using least-squares fitting to estimate the orientation.

### 3.2.1. Image Grouping

The accuracy of localization depends to a large extent on the accuracy of the corresponding points. There are differences between the same road signs in different oblique aerial images due to the variable shooting angles and lighting conditions. Thus, the corre-

sponding points of two identical road signs with completely different views are difficult to extract.

Experiments have shown that there are more corresponding points in road sign images with more similar shooting conditions. Thus, to improve the accuracy of the extracted corresponding points, we first divide the road sign images into several groups. Then, under the geometric constraints placed by the bounding boxes, the corresponding points are searched within the image group to reduce the number of mismatched points.

We group images according to the similarity of shooting angles and lighting conditions, and there are at least two images containing the same road sign in each group.

$$P = \theta(G_1 \cup G_2 \cup \cdots \cup G_i) = \sum \theta(G_i) \tag{5}$$

where $\theta$ denotes the method used to extract corresponding points. $N(G_i)$ is the number of images containing the same road signs, and $N(G_i) \geq 2$.

### 3.2.2. Corresponding Point Extraction with Geometric Constraints

After grouping images, the critical step in stereo vision is to obtain the corresponding points from multiview images. In this research, we take advantage of the SIFT feature to extract the key points in constraint regions defined by the detected bounding box, rather than extracting from the complete oblique aerial image. Then, based on the brute-force matching method, a distance ratio is utilized to optimize the matching of feature points (see Figure 5).

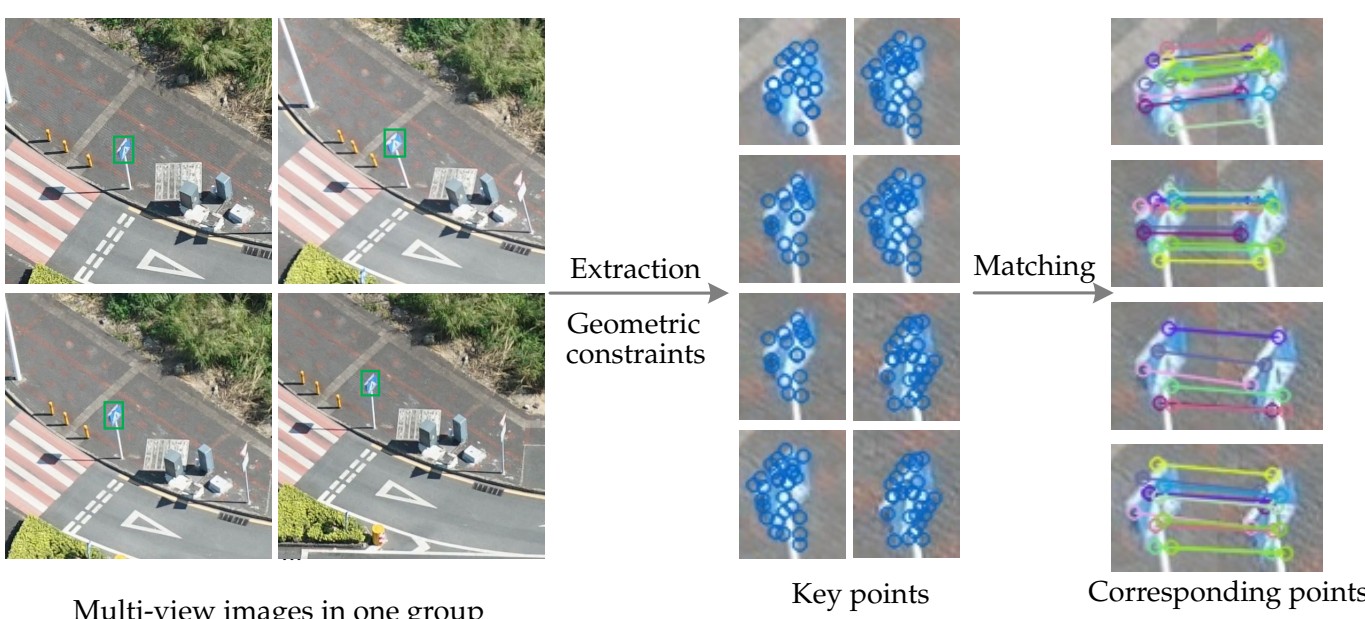

**Figure 5.** Main steps in corresponding point extraction.

The probability of a match being correct can be determined by taking the distance ratio of the closest neighbor to the second closest neighbor. Instead of accepting all the best-matched results generated by the brute-force method, we reject matches in which the distance ratio is more than 0.8. The definition of the distance ratio is provided in Equation (6).

$$D_{ratio} = \frac{D_{closest}}{D_{nextclose}} \tag{6}$$

### 3.2.3. Triangulation and Unrefined Location

The 3D location of any visible object point in space is restricted to the straight line that passes through the center of projection and the projection of the object point. Thus,

the position of a point is at the intersection of the two lines crossing the center of the projection and the projection of the point in each image [58] (see Figure 6). Consider recovering the position of the 3D space point from a pair of point correspondences $V_l \leftrightarrow V_r$ and a pair of camera matrices $P_l$ and $P_r$. Then,

$$V = \tau(V_l, V_r, P_l, P_r) \tag{7}$$

where $\tau$ denotes the triangulation method and $V = (x, y, z, 1)$ is the 3D coordinates of the point.

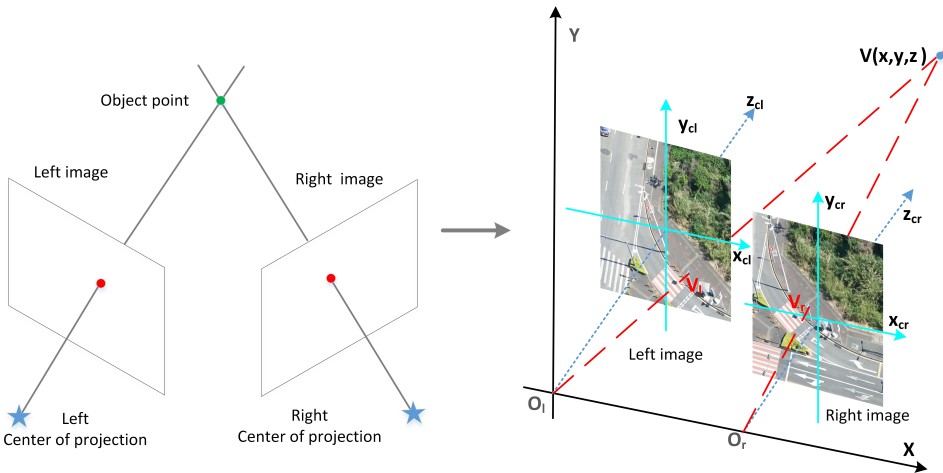

**Figure 6.** Structure from stereo vision. A pair of corresponding points $V_l$ and $V_r$ are used to obtain a 3D point via triangulation.

Triangulation is performed in a single image group, and a pair of corresponding points is used to produce a point in 3D space via triangulation. Then, a set of 3D points is merged from image groups to generate the point cloud that predicts the coarse 3D location of a single road sign. Further optimization of cloud points is necessary to locate and orient accurately in 3D space.

### 3.2.4. Refined 3D Location and Orientation

The 3D points produced from the same road sign within each group of images are merged into a raw point cloud that predicts the coarse 3D location of a single road sign. Such sparse outliers caused by ambiguous corresponding points will lead to localization and orientation errors. Some irregularities can be removed by performing statistical analysis on the neighborhood of each point and trimming those that do not meet a certain criterion.

In this research, sparse outliers are removed using statistical analysis techniques to filter out the noise and avoid misleading information. Specifically, the removal of sparse outliers is based on the computation of the point-to-neighbor distance distribution in the raw point cloud. For each spatial point, we compute the mean distance from that point to each of its neighbors. Assuming that the resulting distribution is Gaussian with a mean $\mu$ and a standard deviation $\sigma$, all points whose mean distances are outside an interval $D_c$ defined by the global distance mean and standard deviation can be considered outliers and removed from the dataset, as illustrated in Equation (8).

$$D_c = \mu + S_{mul} * \sigma, \tag{8}$$

where $S_{mul}$ is a threshold of multiple standard deviations that is set to 1.0 in our experiments.

Regarding 3D location, the refined point cloud is applied to fit a 3D bounding box for each road sign in the space where the side face is vertical to the ground surface while the bottom face is parallel to the ground surface. Figure 7 shows the overall process for refining the road sign 3D location. We use the geometric center point of the fitted 3D bounding box

to clarify the center location of the embedded model. The geometric center point of the fitted 3D bounding box $P_c\ (x_c, y_c, z_c)$ is calculated via Equation (9),

$$P_c(x_c, y_c, z_c) = \begin{cases} x_c = x_{min} + \frac{1}{2}(x_{max} - x_{min})) \\ y_c = y_{min} + \frac{1}{2}(y_{max} - y_{min})), \\ z_c = z_{min} + \frac{1}{2}(z_{max} - z_{min})) \end{cases} \tag{9}$$

where $x_{min}$, $y_{min}$ and $z_{min}$ are the minimum x, y and z of the 3D points, respectively, and $x_{max}$, $y_{max}$ and $z_{ma}x$ are the maximum x, y and z of the 3D points.

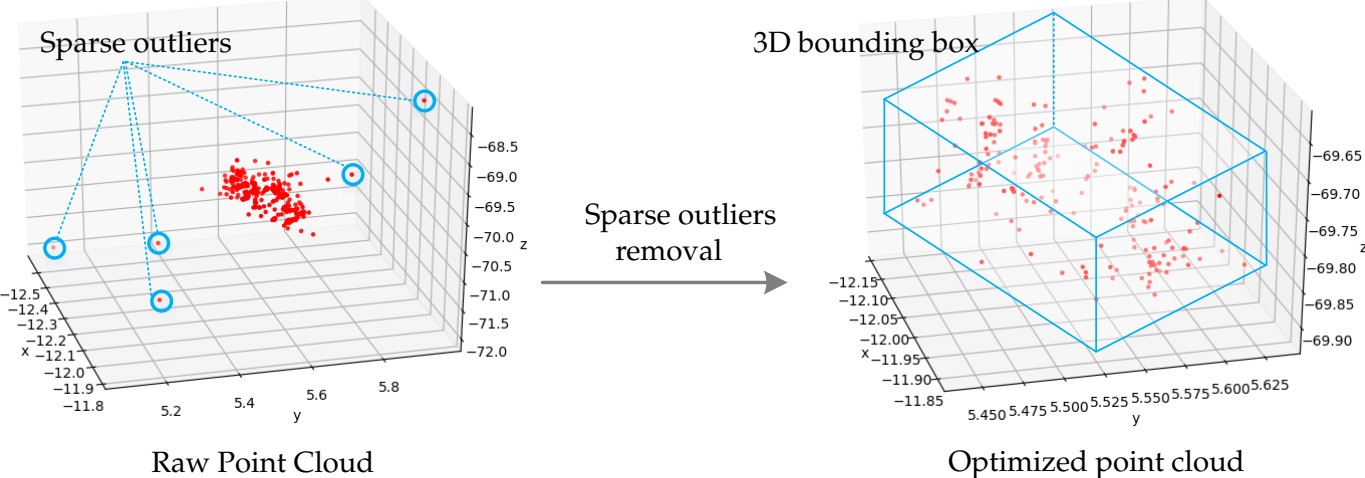

**Figure 7.** Road sign 3D localization refining. Sparse outliers (blue circles) are removed via the statistical analysis method. The 3D bounding box (blue box) is fitted based on the refined point cloud.

Concerning orientation, we fit a plane to the refined point cloud using least-squares fitting to indicate the orientation of a road sign. Generally, the road sign face should be vertical to the ground surface. We fit the plane under the condition that the fitted plane is perpendicular to the ground surface. Thus, the normal vector of the fitted plane is perpendicular to the normal vector of the ground surface.

The plane we want to estimate is $z = ax + by + c$, where $a$, $b$ and $c$ are scalars. This defines a plane that best fits the samples in the sense that the sum of the squared errors between $z_i$ and the plane values $ax_i + by_i + c$ is minimized. We can define the error function for the least-squares minimization as Equation (10).

$$E_0(a, b, c) = \sum_{i=1}^{n}[(ax_i + by_i + c) - z_i]^2 \tag{10}$$

under the condition of Equation (11)

$$v_1 \perp v_2 \rightarrow v_1 \cdot v_2 = 0, \tag{11}$$

where $v_1$ is the normal vector to the ground surface and $v_2$ is the normal vector to the fitted plane. $v_1$ is perpendicular to $v_2$.

Thus, we utilize a point $P_c(x_c, y_c, z_c)$ and the vectors $v_1$ and $v_3(a, b, -1)$ to clarify the position and orientation of the embedded road sign model, where $P_c$ is the center point of the fitted 3D bounding box, $v_1$ is the normal vector to the ground surface and $v_3$ is the normal vector to the fitted plane.

*3.3. Road Sign Model Matching and Embedding*

There are general guidelines to follow in the design of road signs to conform to basic standards. Thus, following the same rules, 3D templates and texture databases of road signs are developed for template fitting and texture matching, respectively.

In this research, we create a sketch model and texture database of road signs for model matching. Considering the functionality and shape of road signs, we classify them into three categories: warning signs with a triangular shape, prohibitory signs with a circular shape and mandatory signs with a square shape.

In general, prohibitory traffic signs are used to prohibit certain types of maneuvers or some types of traffic. Such signs have round shapes with radii of 30, 40 or 50 cm. Warning signs are equilateral triangles with sides lengths of 70, 90 or 110 cm, mostly bright yellow in color. Mandatory signs are used to define the obligations of all traffic that uses a specific area of the road, generally using a white safety symbol on a blue background. The standard sizes of mandatory signs include $60 \times 60$ and $80 \times 80$ cm. The computer-aided design 3D templates and texture database are clarified in Figure 8.

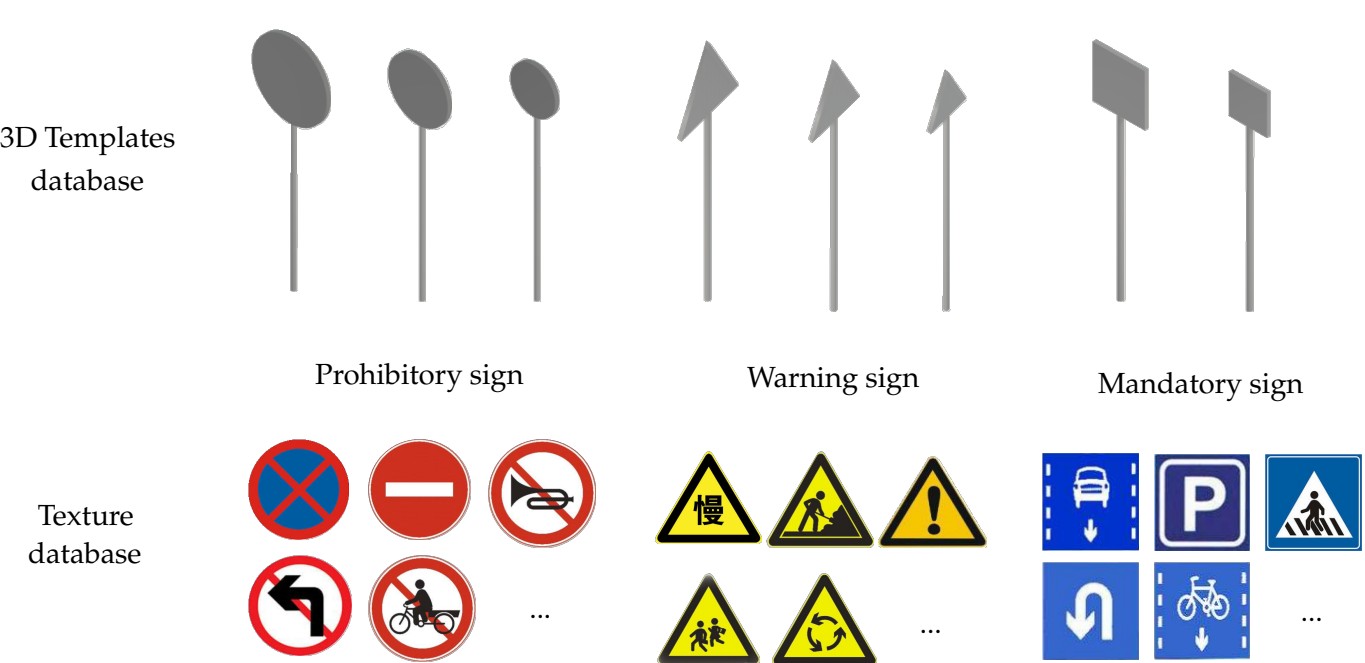

**Figure 8.** 3D templates and texture database of road signs.

On the basis of the classification and 3D bounding box results, the correspondence template and texture are retrieved from the database to generate a rendered road sign model, and the template with the size closest to the generated 3D bounding box is selected. Meanwhile, the correspondence texture is selected to render the selected template according to the similarity of the image. The schema of road sign model generation is illustrated in Figure 9. Once we obtain a complete computer-aided design road sign from the database, we position it in the 3D urban scene based on the estimated 3D location and orientation: the embedded road sign model is placed upright to the ground surface.

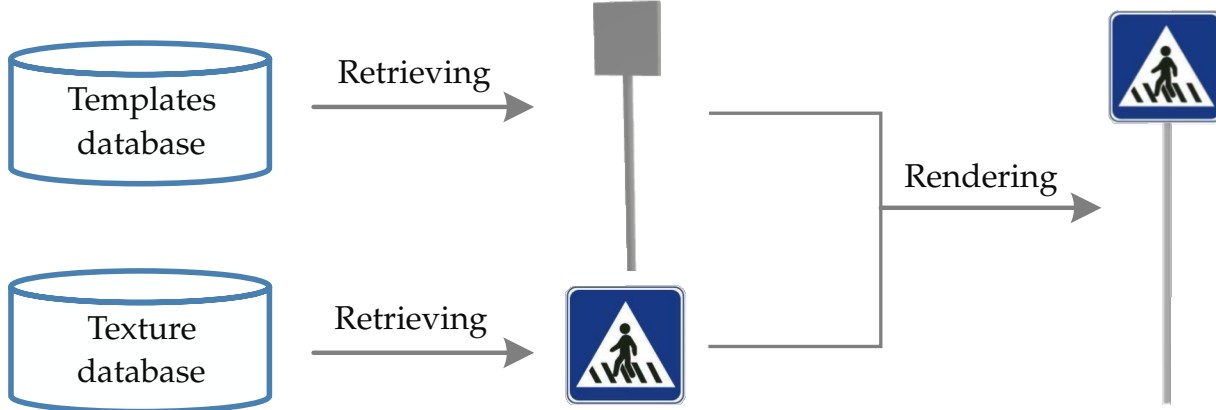

**Figure 9.** Schematic of road sign model generation.

*3.4. Evaluation Metric*

Average precision (AP) is a popular measure of the accuracy of object detectors, and a general definition of AP is the area under the precision-recall curve. Mean average precision (mAP) is the average of the AP. Equation (12) is used to calculate the average precision for each road sign type.

$$AP = \sum_n (R_n - R_{n-1})P_n \tag{12}$$

where $P_n$ and $R_n$ are recall and precision. Precision represents the correctness of the predictions, while recall measures the ability of the detector to identify all positive samples. Mathematical definitions of precision and recall are given in Equations (13) and (14).

$$Precision = \frac{TP}{TP + FP} \tag{13}$$

$$Recall = \frac{TP}{TP + FN} \tag{14}$$

where *FP* and *TP* are the numbers false positives and true positives, respectively, and *FN* is the number of false negatives.

In our experiments, we use IoU with a predefined threshold of 0.5 to classify whether the prediction is a true positive or a false positive. The IoU measures how much our predicted boundary overlaps with the ground truth (the real object boundary). An IoU less than 0.5 indicates an FP; otherwise, the result is a TP. Equation (15) illustrates how to calculate the IoU.

$$IoU = \frac{area\ of\ overlap}{aera\ of\ union} \tag{15}$$

## 4. Experiments

*4.1. Aerial Oblique Image Collection and Preprocessing*

Flight planning should consider the best approach to cover a whole object with the minimum flight time, visibility of vertical structures, occlusions due to context, acquisition of all parts of the objects (the closest and the farthest) with similar resolution and suitable camera inclination [59]. Considering these problems, we collect oblique aerial images via the Shuangyu 5.0B data collection platform carried on a DJI M600 UAV, which deploys two SONY RX100 M2 cameras with 42 million camera pixels. We collect oblique aerial images in Shenzhen area, China.

Oblique aerial images are processed by aerial triangulation to correct distortion and export stereo camera parameters with the commercial software ContextCapture. The process of oblique aerial image collection, preprocessing and synthesis is depicted in Figure 10.

Considering the memory limits of computer hardware, undistorted oblique aerial images with a size of 7952 × 5304 pixels are cropped to 1024 × 1024 pixels for CNN model training. Leaving out those images without road signs, we select 450 original images for data synthesis and augmentation. Figure 10 shows the process of oblique aerial image collection, preprocessing, and synthesis.

The number of images containing each road sign category and the number road signs in each category before and after data synthesis are shown in Figure 11. Before data synthesis, these numbers are small. The number of images with mandatory signs is nearly six times that of warning signs, while the number of mandatory signs in images is more than seven times that of warning signs. Thus, to ensure that the image number and object number of each category were as balanced as possible, we used the proposed data synthesis method to augment the training dataset.

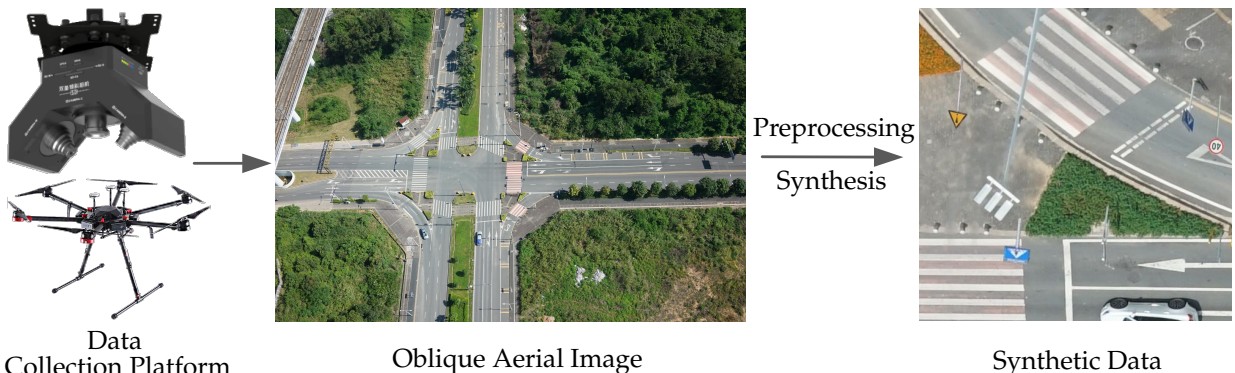

**Figure 10.** The process of oblique aerial image collection, preprocessing and synthesis.

We split the training dataset into two groups by randomly selecting 70% of the images for training and the remaining 30% for testing and altered the obtained images 24 times via rotation, mirror transformation and gamma transformation to avoid overfitting. Original and synthetic training data are presented in Figure 12. The final set includes 450 images with a pixel size of 1024 × 1024.

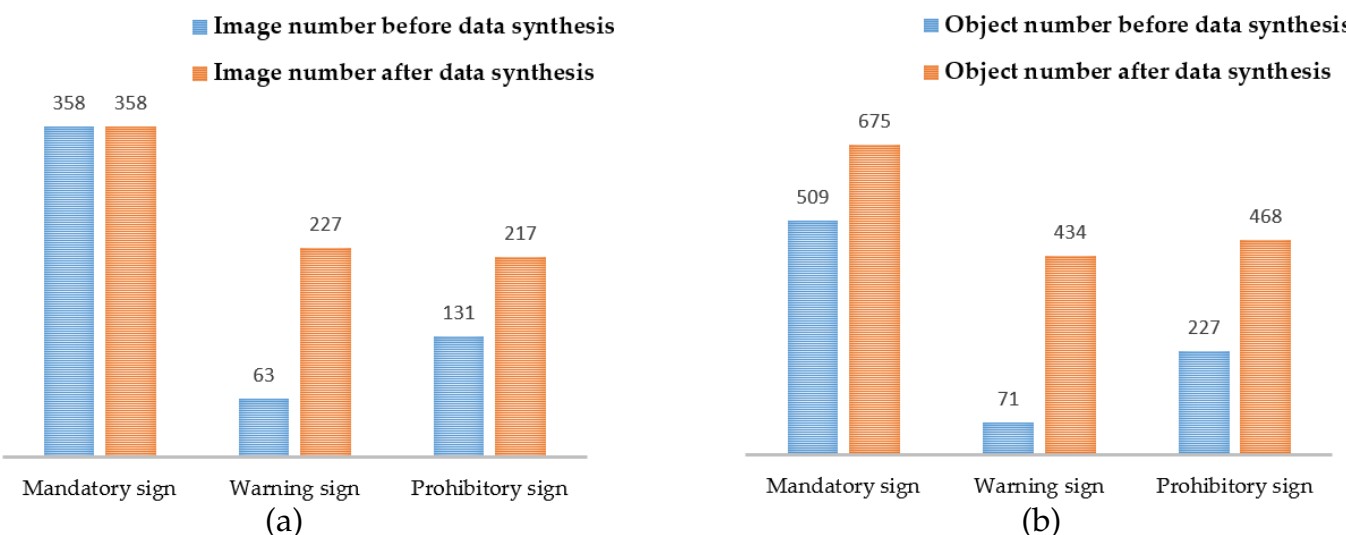

**Figure 11.** (**a**) The number of oblique aerial images containing each road sign category before and after data synthesis. (**b**) The number of road signs belonging to each category before and after data synthesis.

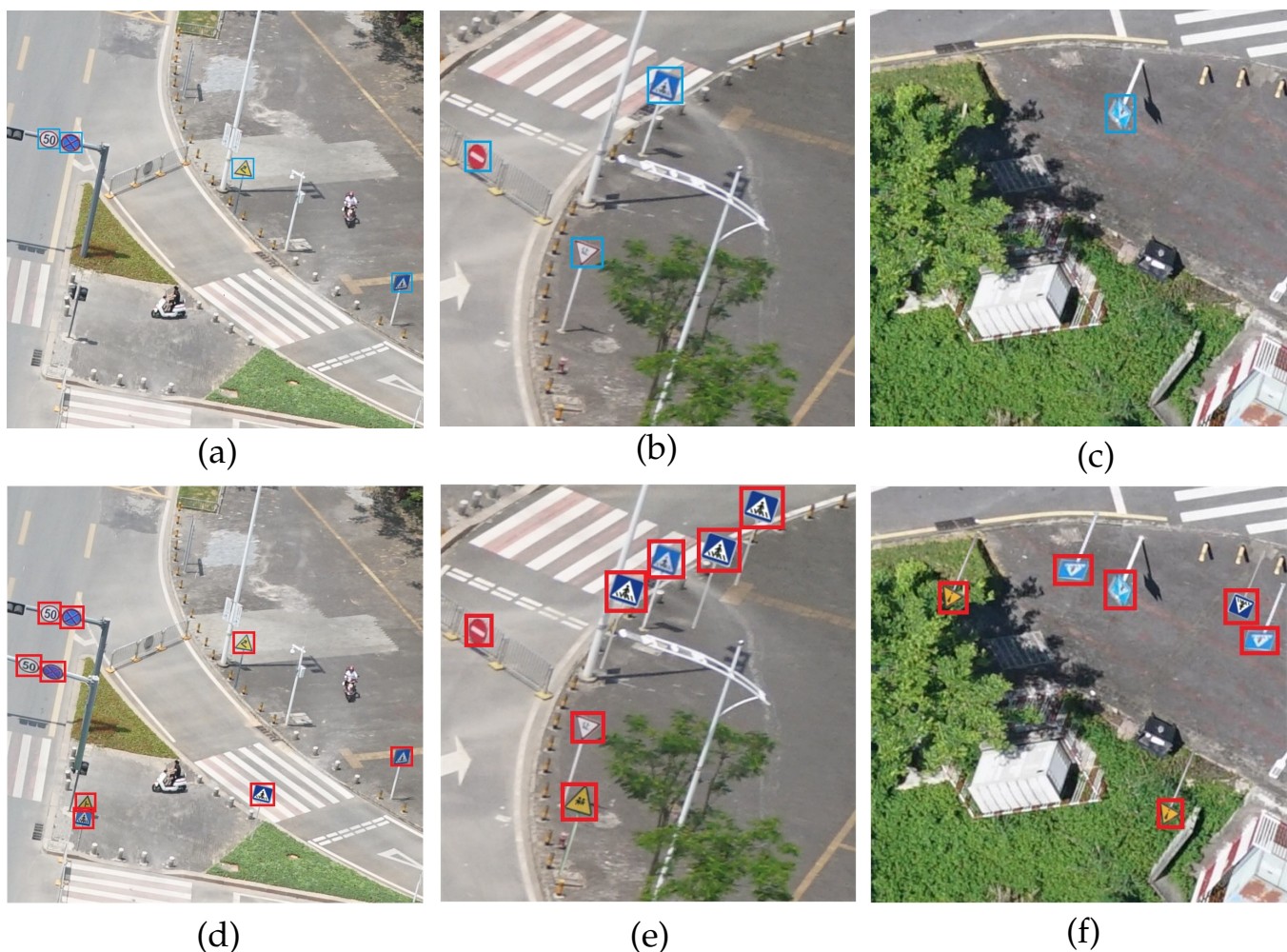

**Figure 12.** Comparison of training data before and after synthesis and augmentation. (**a**–**c**) Original oblique images, and (**d**–**f**) synthetic images. We cropped the training images to a pixel size of 512 × 512 in this figure.

*4.2. Road Sign Detection*

4.2.1. CNN Model Training Setup

For detector training, we implement experiments in the PyTorch framework with the Ubuntu 16.04 operating system. The computer hardware is an RTX2080ti GPU with 11 GB memory. We train the detector for ten epochs with an initial learning rate of 0.001 and decrease the learning rate at the 5th and 8th epochs. We use a weight decay of 0.0001 and momentum of 0.9. The minibatch sizes of the RPN stage and classification stage are 256 and 256. Furthermore, ResNet101, which is pretrained on the ImageNet dataset [60], is used to extract features in this paper.

Comparative experiments are conducted with the baseline of faster R-CNN to verify the effectiveness of the proposed method. We first train faster R-CNN with the original data and synthetic dataset. Then, we implement guided anchoring (GA) with synthetic data, balanced learning (BL) and the proposed combined method (BL and GA combined) based on faster R-CNN. We consider anchors with an IoU overlap below 0.3 as negative samples and those with an IoU overlap greater than 0.7 as positive samples for all experiments. Regarding the methods that require predefined anchors, the size of the anchors and the ratio of anchors are set to [4, 8, 16, 32, 64] and [0.5, 1.0, 2.0], respectively.

### 4.2.2. Detection Result

The detection results based on the proposed method are shown in Figure 13. Green bounding boxes represent TPs, while red bounding boxes represent FPs or FNs. Most road signs are distinguished from other objects and located with well-fitted bounding boxes.

The results of comparative experiments are shown in Table 2, including the results of faster R-CNN (Faster) and guided anchoring (GA) using the original data, GA using synthetic dataset (GA&SD), balanced learning (BL), and the proposed method that combined BL with GA (BL&GA). All the experiments are based on the backbone of ResNet-101-FPN.

**Table 2.** Detection results of the comparison approaches.

| Method | Mandatory Sign | Warning Sign | Prohibitory Sign | mAP |
|:---:|:---:|:---:|:---:|:---:|
| Faster | 0.803 | 0.785 | 0.909 | 0.832 |
| Faster with GA | 0.876 | 0.832 | 0.818 | 0.842 |
| Faster with SD | 0.907 | 0.886 | 0.897 | 0.897 |
| Faster with GA&SD | 0.907 | 0.908 | 0.904 | 0.906 |
| Faster with BL | 0.908 | 0.952 | 0.903 | 0.921 |
| Faster with BL&GA | 0.909 | 0.987 | 0.909 | **0.935** |

Faster: faster R-CNN method. Faster with GA: training the faster R-CNN model with guided anchoring. Faster with SD: training the faster R-CNN model using synthetic data. Faster with GA&SD: training the faster R-CNN model with guided anchoring and synthetic data. Faster with BL: training the faster R-CNN model with balanced learning. Faster with BL&GA: training the faster R-CNN model with balanced learning and guided anchoring.

Compared to that of faster R-CNN, the proposed method for synthesizing data improves the mAP from 83.2% to 89.7%. The synthetic data method enhances the mAP by 6%. The guided anchoring strategy contributes slightly, with a 1% enhancement to the mAP. However, the guided anchoring strategy improves the mAP by 7.4% when using synthetic data. The balanced learning method improves the mAP from 83.2% to 92.1%, and the combined balanced learning and guided anchoring method outperforms the other methods in our experiments, improving the mAP by 10.3%.

In comparison, synthetic data effectively improves the detection accuracy: the performance of the detector improves as the number of road signs increases. However, the accuracy is not enhanced further after the object number exceeds a certain threshold. Both the guided anchoring strategy and balanced learning method contribute to mAP enhancement, and the proposed method obtained the best mAP of 93.5%.

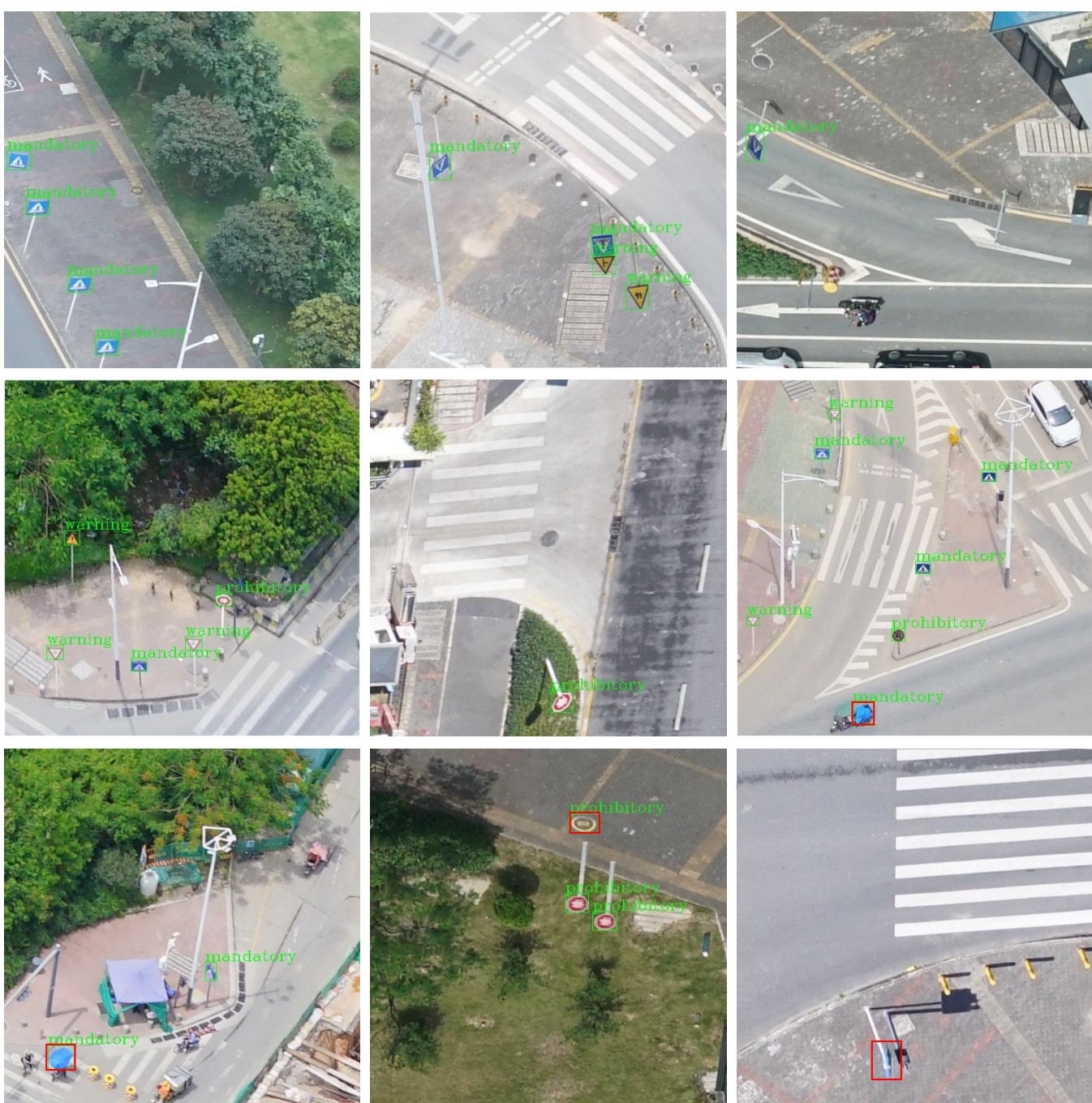

**Figure 13.** Road sign detection result in cropped oblique aerial images. Green bounding boxes represent TPs, while red boxes represent FPs or FNs. We cropped the images of the detection results to a pixel size of 512 × 512 in this figure. The proposed method obtained an mAP of 93.5%.

### 4.3. Road Sign Embedded Modeling

We apply the proposed method for embedded modeling of road signs in the area of Shenzhen, China. Generally, road signs are located along streets and are easy to find at junctions. Several embedded modeling cases are shown to illustrate the feasibility of the proposed method.

#### 4.3.1. Extracting Corresponding Points with Geometric Constraints in Image Groups

In our experiment, the SIFT feature and optimized brute-force matching method are utilized for key point extraction and matching. However, the same road signs in different

oblique aerial images may vary substantially due to variations in shooting angles and lighting conditions. Thus, it is not easy to extract corresponding points from the same two road signs with different views. Figure 14 shows the matching results of road signs with different and similar shooting angles.

The results of corresponding point extraction in Figure 14 show that road sign images with more similar shooting angles have more corresponding points with higher accuracy. Distortion and lighting changes in images impact the corresponding point extraction. Although incorrect corresponding points remain, most of the errors are avoided by grouping.

Therefore, we group images containing road signs based on the similarity of the shooting angle and lighting conditions. At least two images in each group contain the same road sign. Then, corresponding points are extracted with geometric constraints in the image groups. Additionally, instead of using all the best-matched point pairs, we reject all matches in which the distance ratio is more than 0.8 to further improve the accuracy of the corresponding points. Finally, the corresponding points of all the groups are merged to generate a coarse 3D location.

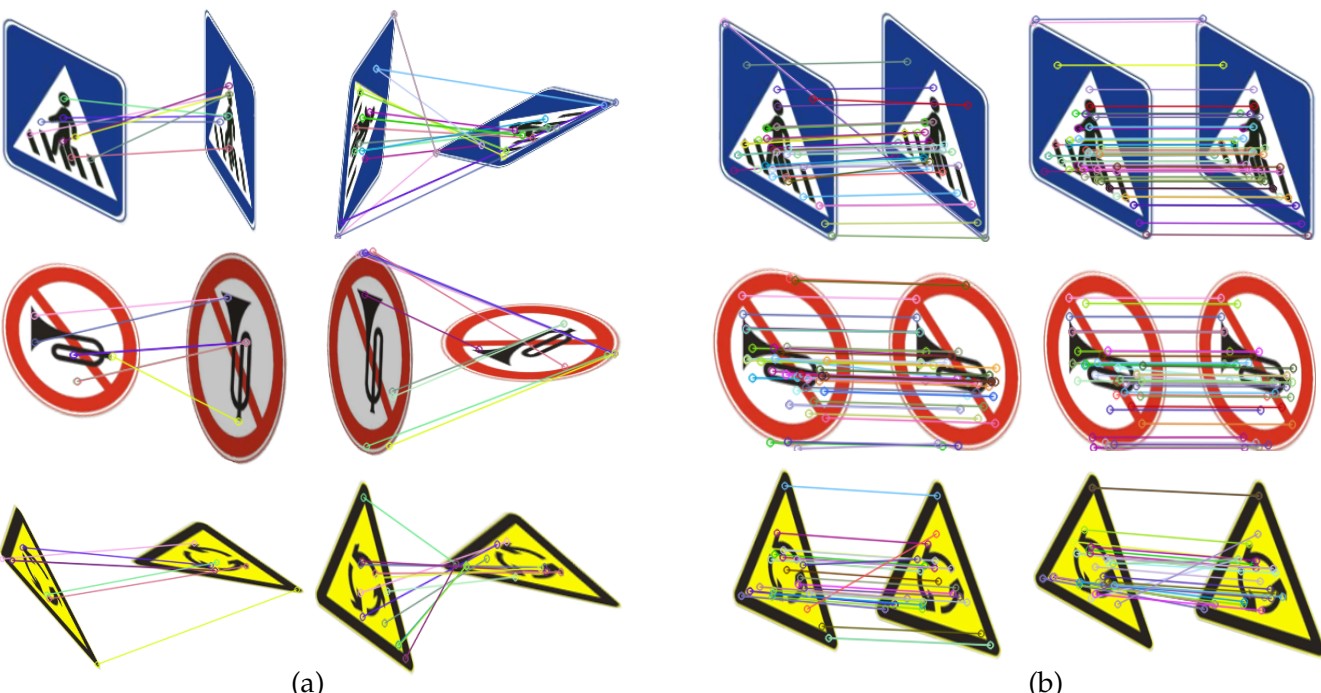

(a)  (b)

**Figure 14.** Corresponding points: grouping images before matching. (**a**) Corresponding points extracted from road signs with different shooting angles. (**b**) Corresponding points extracted from road signs with similar shooting angles.

### 4.3.2. Localization and Orientation

As mentioned in Section 3.2.4, we adopt statistical outlier removal to improve the accuracy of the corresponding points. In this research, we set the number of adjacent points to 50 and the value of $S_{mul}$ to 1.0, thereby preserving 75% of the spatial points of the raw point cloud in the experiments. The refined point cloud is then used for more accurate 3D localization and orientation. Based on the spatial information of the ground surface, the geometric center of the 3D bounding box is the center of the embedded model. The fitted plane indicates the orientation. Four cases, *a*, *b*, *c*, and *d*, are depicted in Figure 15.

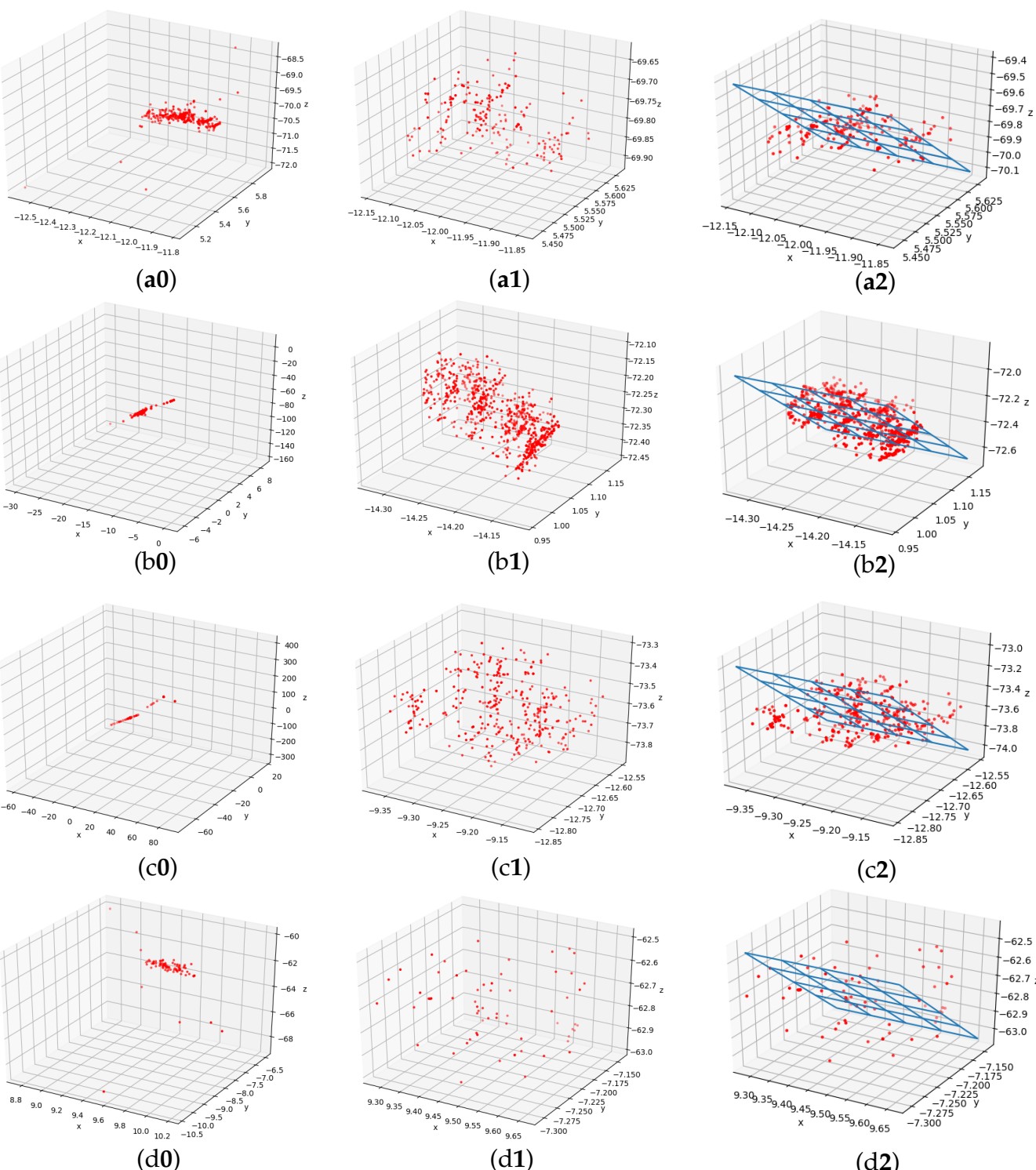

**Figure 15.** (**a0–d2**) The first column is the result of triangulation. The second column is the refined point cloud. Removing outliers to refine 3D localization, the number of adjacent points is set to 50, and the value of Smul is set to 1.0, preserving 75 spatial points of the raw point cloud. The third column is the fitted plane using least-squares fitting with the condition that the fitted plane is vertical to the ground surface.

4.3.3. Model Embedding

For model embedding, we retrieve the best-matched road sign template and texture from the predefined database to obtain a complete road sign model and then lay it on the

urban scene based on the predicted 3D position and orientation. Generally, the embedded road sign model is supposed to be upright to the ground surface. Regarding the 3D location, we consider the geometric center of the 3D bounding box as the center of the computer-aided design 3D road sign model. An estimated normal vector of the fitted plane and a normal vector of the ground surface are applied to orient the computer-aided design road sign model. We selected several cases to demonstrate the results of embedding modeling, revealing the feasibility of the proposed method. The results of embedded modeling in the experimental area are shown in Figure 16.

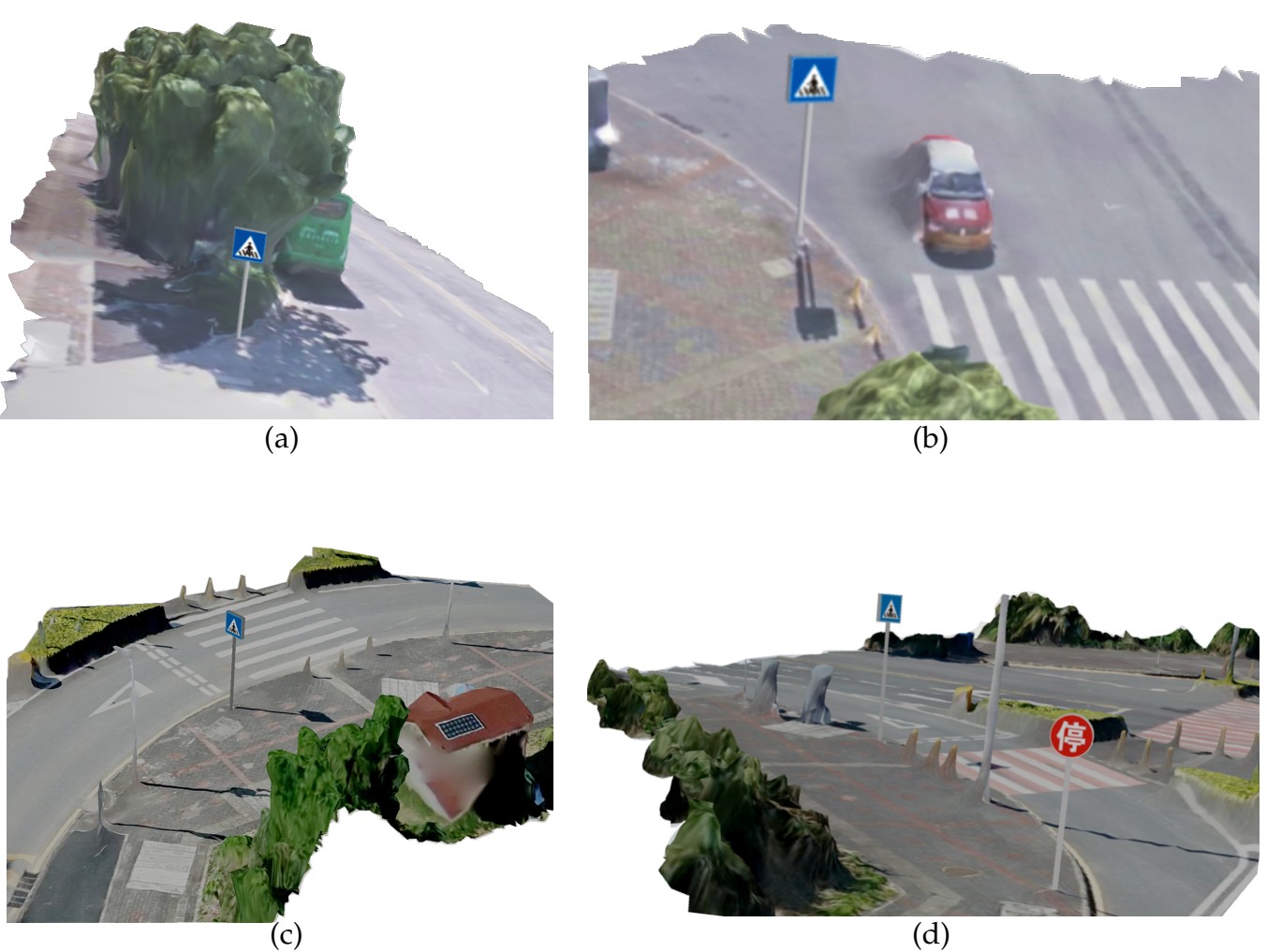

(a)

(b)

(c)

(d)

**Figure 16.** (**a**–**d**) Several cases of road sign embedded modeling results.

## 5. Discussion

We present a pipeline for embedding road sign models based on a deep CNN in this paper. There are three main steps in the proposed method: detecting road signs from oblique aerial images, locating and orienting road signs in 3D scenes, and retrieving templates and texture to embed computer-aided design models of road signs. The advantages and efficiencies of the proposed algorithm include two main aspects. First, we present an end-to-end balanced-learning framework for small object detection that takes advantage of the region-based CNN and a data synthesis strategy. Second, corresponding points applied for triangulation are extracted with the geometric and image group constraints and optimized via a statistical method.

The performance of the detector improves as the number of road signs increases. However, accuracy is not further enhanced after the object number exceeds a certain

threshold. Although data synthesis improves the detection performance, we did not analyze the relationship between the number of objects and the accuracy of the detection method in detail.

Concerning the detection approach, we take advantage of the region-based CNN and a data synthesis strategy to improve the accuracy. The proposed method achieves a high mAP for road sign detection, yet there remains room for improvement in accuracy. Additionally, it is not sufficient to roughly classify all road signs into three categories according to shape and functionality.

Regarding 3D localization and orientation, we extract corresponding points with geometric constraints in image groups, and we exploit a statistical analysis method to refine the results of triangulation. However, in our experiments, we use only the SIFT feature and the optimized brute-force matcher to extract corresponding points such that other methods may overwhelm the implemented method. Although we use strategies such as image grouping, the distance ratio, and statistical analysis for optimization, errors remain.

## 6. Conclusions

In this paper, we presented a pipeline for embedding road sign models based on deep convolutional neural networks (CNNs). The experimental results showed that the proposed method obtained an mAP of 93.5% in road sign detection and produced visually plausible embedded results. Moreover, the proposed method is effective for road sign modeling in oblique photogrammetry-based 3D scene reconstruction. The proposed method also provided a reference for modeling some specific objects, such as street lamps and billboards, in oblique photogrammetry-based 3D scenes. In future work, we will enrich the diversity of oblique aerial image datasets and implement more detailed classification criteria. Meanwhile, we will develop better feature extractors and matchers with fewer errors.

**Author Contributions:** Z.M. and X.H. contributed equally to design this study and write this manuscript, including the figures and tables. F.Z., Q.Z., X.J. and Y.G. contributed significantly to the discussion of results and manuscript refinement. All authors have read and agreed to the published version of the manuscript.

**Funding:** This research was funded by the National Key R&D Program of China No. 2020YFC1523003 and the Open Research Fund Program of Shenzhen Key Laboratory of Spatial Smart Sensing and Services No. SZU51029202009.

**Data Availability Statement:** The data presented in this study are available on request from the corresponding author.

**Acknowledgments:** The authors would like to thank the researchers who have provided the open-source MMDetection toolbox, SIFT algorithms and other open-source software, which have been extremely helpful to the research in this paper.

**Conflicts of Interest:** The authors declare no conflicts of interest.

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
