# Peer review of "Deep Neural Networks for Road Sign Detection and Embedded Modeling Using Oblique Aerial Images"

_remotesensing, doi:10.3390/rs13050879_

Round 1
Reviewer 1 Report
The paper presents an interesting method to detect road signs using convolutional neural networks and aerial images. It is an interesting topic and has a well-structured method. I suggest accepting the paper after major revisions.
- Abstract: What does it mean mAP?
- Introduction: Why did you select CNN? Are there other algorithms in the literature for this scope?
- Line 181: "Generally, we present an end-to-end balance-learning framework for small object
182 detection, which takes advantage of the region-based CNN and a data synthesis strategy, see Figure ??" --> You need to define the number of the figure. Check the whole document as I found this issue many times in the document (i.e. L415, L433). - Line 276: Why did you decrease the epochs?
- Line 377: How did you define the weight decay and the momentum?
- Section 4.2.2: needs to move Section 3 as it describes a tool to assess your algorithm.
- Table 2: it would be useful to have the abbreviations under the table.
- How much time does the algorithm need? Is this an aspect that makes your algorithm to outperform other algorithms?
- What about signs in poor condition? Is it feasible to be recognised?
- From which area did you capture data? Would it possible to use this algorithm in various countries or continent?
Author Response
Dear reviewer,
Thanks for your comments concerning our manuscript entitled “Deep Neural Networks for Road Sign Detection and Embedded Modeling Using Oblique Aerial Images” (remotesensing-1085488)
Those comments are helpful for revising and improving our paper. We have studied the comments seriously and made corrections carefully. The responses to your comments are marked in red, and the corrections in the newly submitted version are marked in blue and line number. The main corrections in the paper and the responses to your comments are shown.
We appreciate your warm work earnestly, and hope that the correction will meet with approval.
Kind regards,
Zhu Mao on behalf of the authors

Reviewer 2 Report
The work is interesting and presents clear merits, but there is a need for supplementary explanations and clarifications by the authors to make the article publishable. So I have requested some modifications and changes in this direction.
Well done. The work is interesting and presents clear merits, but needs some things to be reconsidered:
- the analytic procedure needs to be explained more in detail and generally clarified.
- What types of outliers do you have found in your analysis? What method have you used to detect these outliers?
Author Response
Dear reviewer,
Thanks for your comments concerning our manuscript entitled “Deep Neural Networks for Road Sign Detection and Embedded Modeling Using Oblique Aerial Images” (remotesensing-1085488).
Those comments are helpful for revising and improving our paper. We have studied the comments seriously and made corrections carefully. The responses to your comments are marked in red, and the corrections in the newly submitted version are marked in blue and line number. The main corrections in the paper and the responses to your comments are shown.
We appreciate your warm work earnestly, and hope that the correction will meet with approval.
Kind regards,
Zhu Mao on behalf of the authors

Reviewer 3 Report
A review of this paper on its scientific merits is very difficult due to the numerous issues in English usage, descriptions that use terminology that is not defined, and the absence of nearly all of the referenced figures. The attached annotated manuscript provides highlighted examples of English usage (only on the first few pages), some examples of undefined terminology, and missing figures. This paper should only be reconsidered after extensive editing to enable a systematic review of the problem definition, approach taken to address the problem, and the results obtained.

Author Response
Dear reviewer,
Thanks for your comments concerning our manuscript entitled “Deep Neural Networks for Road Sign Detection and Embedded Modeling Using Oblique Aerial Images” (remotesensing-1085488).
Those comments are helpful for revising and improving our paper. We have studied the comments seriously and made corrections carefully. The responses to your comments are marked in red, and the corrections in the newly submitted version are marked in blue and line number. Please see the attachment. The main corrections in the paper and the responses to your comments are shown.
We appreciate your warm work earnestly, and hope that the correction will meet with approval.
Kind regards,
Zhu Mao on behalf of the authors

Round 2
Reviewer 1 Report
The authors have addressed satisfyingly my comments. I suggest accepting the paper in the current form.
Reviewer 3 Report
This version of the paper is a significant improvement over the previous submission and successfully addresses my previous comments.